# Generation of gravity waves from thermal tides in the Venus atmosphere

Norihiko Sugimoto [1,2✉], Yukiko Fujisawa[2], Hiroki Kashimura [3,4], Katsuyuki Noguchi[5], Takeshi Kuroda[6], Masahiro Takagi [7] & Yoshi-Yuki Hayashi [3,4]

Gravity waves play essential roles in the terrestrial atmosphere because they propagate far from source regions and transport momentum and energy globally. Gravity waves are also observed in the Venus atmosphere, but their characteristics have been poorly understood. Here we demonstrate activities of small-scale gravity waves using a high-resolution Venus general circulation model with less than 20 and 0.25 km in the horizontal and vertical grid intervals, respectively. We find spontaneous gravity wave radiation from nearly balanced flows. In the upper cloud layer (~70 km), the thermal tides in the super-rotation are primary sources of small-scale gravity waves in the low-latitudes. Baroclinic/barotropic waves are also essential sources in the mid- and high-latitudes. The small-scale gravity waves affect the three-dimensional structure of the super-rotation and contribute to material mixing through their breaking processes. They propagate vertically and transport momentum globally, which decelerates the super-rotation in the upper cloud layer (~70 km) and accelerates it above ~80 km.

[1] Department of Physics, Keio University, Yokohama, Japan. [2] Research and Education Center for Natural Sciences, Keio University, Yokohama, Japan. [3] Department of Planetology, Kobe University, Kobe, Japan. [4] Center for Planetary Science, Kobe University, Kobe, Japan. [5] Faculty of Science, Nara Women's University, Nara, Japan. [6] Department of Geophysics, Tohoku University, Sendai, Japan. [7] Faculty of Science, Kyoto Sangyo University, Kyoto, Japan. ✉email: nori@phys-h.keio.ac.jp

In the terrestrial atmosphere, gravity waves generated in the troposphere—most of which are orographically forced waves and non-orographic waves associated with convection—drive the general circulation of the middle atmosphere[1]. The non-orographic gravity waves are also generated spontaneously from jet/front systems[2], although the large-scale motions are almost balanced[3,4]. Since small-scale gravity waves appear at jet-exit regions, they are often called Jet-Exit Region Emitted (JEREmi) waves[2]. The spontaneous generation of gravity waves for the terrestrial atmosphere has traditionally been investigated in horizontally two-dimensional shallow water system[5–7] (usually called Lighthill–Ford spontaneous adjustment), and extended to three-dimensional models[8,9]. Theoretically, two mechanisms are possible to explain small-scale JEREmi waves[10]. One is the velocity-variation mechanism in which strong divergence/convergence of the flow generates vertical motions, which lead to gravity waves. The other is the mountain-wave-like mechanism in which the deformation of isentropic surfaces acts as a mountain; the flow over isentropic surfaces produces upward/downward motions of gravity waves.

The gravity waves are also expected to play essential roles in the Venus atmosphere[11]. It has been suggested using idealistic vertical convection models that the gravity waves could be generated by convective motions in the cloud layer[12–14], and by cloud feedback radiative heating[15]. Radio occultation measurements of the Venus Express[16] and Akatsuki observed temperature perturbations, indicating gravity waves with vertical wavelengths less than 4 km. While these temperature perturbations are eminent around the cloud layer (~60 km), their magnitudes decrease around the cloud-top (~70 km) and increase above the cloud-top (~80 km), suggesting another source of gravity waves other than the cloud layer convection or cloud feedback radiative heating. Visible and Infra-red Thermal Imaging Spectrometer-Mapper (VIRTIS-M) onboard the Venus Express also suggested that small-scale gravity waves with horizontal wavelengths in the range of 90–400 km exist in the upper atmosphere in the range of 110–140 km altitudes[17]. Further, at the cloud tops (62–70 km), many types of small-scale gravity wave whose horizontal wavelengths are tens of kilometres were observed by the Venus Monitoring Camera[18,19] (VMC) and VIRTIS-M[20] onboard the Venus Express. Following the experiences on the terrestrial atmosphere, possible effects of those gravity waves have also been incorporated as parameterised schemes in general circulation models (GCMs) for the Venus thermosphere[21,22]. Nevertheless, the characteristics of gravity waves in the Venus atmosphere, such as their generation, propagation, and geographical distribution are poorly understood. Especially, the spontaneous gravity wave radiation from the super-rotation has not been investigated because the resolutions of the typical Venus GCMs so far have been, for instance, around T10 or T21[23] (triangular truncation at wavenumber 10 or 21 with 32 × 16 or 64 × 32 horizontal grids, respectively), which is insufficient to resolve the abovementioned issues. Recently, it has been highlighted that the medium-scale gravity waves appear in moderate resolution Venus GCMs[24,25] (e.g., T63 with 192 × 96 horizontal grids). However, the details remain unclear due to the lack of resolution in previous studies.

Recently, we developed a Venus GCM modified from an Atmospheric GCM optimised For the Earth Simulator (AFES)[26] and named AFES-Venus for a Venus version of AFES. Starting from an idealised super-rotating flow, AFES-Venus reproduced baroclinic waves[27], the super-rotation with planetary-scale waves[24,28] and thermal tides[29] under the realistic distributions of solar heating and static stability, polar and equatorial temperature distributions consistent with observation[30,31] and planetary-scale streak structure[32].

In this study, we resolve small-scale gravity waves and investigate the spontaneous gravity wave radiation in high-resolution simulations of AFES-Venus. The impact of thermal tides, which significantly affects the super-rotation[33,34], on small-scale gravity waves is examined by conducting two experiments with and without the thermal tides. The atmospheric motions are driven by the solar heating with and without the diurnal component in the nominal and Qz (in which only the zonal component [i.e., latitudinal dependence] of the solar heating Q is considered) cases, respectively. Since the thermal tides are excited by the diurnal component of the solar heating[35,36], they are excluded in the Qz case. Our main goal is to demonstrate that gravity waves are spontaneously radiated from the thermal tides in the super-rotation and their activities in the atmosphere are not small.

## Results

**Distribution of gravity waves.** In the nominal case (Supplementary Fig. 1a), super-rotation with the equatorial wind speed of more than 100 m s$^{-1}$ and with weak mid-latitude jets at the cloud-top (~70 km) level reached a quasi-equilibrium state, which was similar to those reported in previous Venus GCM studies[24,25]. For Qz, because equatorward momentum transport due to the thermal tides disappeared, the equatorial wind speed decreased and the mid-latitudes jets were intensified near the cloud-top compared with the nominal case (Supplementary Fig. 1b).

Figure 1a, b show that small-scale gravity waves with horizontal wavelengths of ~250 km were resolved at the cloud-top level in the high-resolution experiments (with ~20 km horizontal grid intervals). As shown in Fig. 1a, the horizontal distribution in the nominal case was related to the planetary-scale thermal tides, which were locked to the solar motion (see Supplementary Movie 1).

At the equator, these waves seemed to be generated from the edge of the large-scale pressure deviations associated with the thermal tides, and propagate vertically (Fig. 1c). For Qz, contrastingly, the gravity waves in the low-latitudes were insignificant (Fig. 1d). It is strongly suggested that the small-scale gravity waves in the low-latitudes would be radiated from the thermal tides (see Supplementary Movies 3 and 4). In addition, the small-scale gravity waves appeared mainly in the mid-to-high-latitudes, and they were advected by the super-rotation (Fig. 1b and Supplementary Movie 2). It is inferred that barotropic/baroclinic waves, which were active in the mid-latitudes and high-latitudes, were the source of these waves. The gravity waves in the high-latitudes were more active for Qz than for the nominal because the mid-latitudes jets for Qz were shifted poleward and were stronger than for the nominal (see Supplementary Fig. 1b). The emission of gravity waves from the lower cloud layer (~50 km, below the lower boundary of Fig. 1c, d) seemed to occur neither the nominal nor Qz, which is because the implemented radiative cooling was a simple Newtonian type that forced atmospheric temperature to relax towards the prescribed mean profile, whose static stability was small but stable in the cloud layer. There was little chance for the model to activate the convective adjustment scheme or some crude representations of vertical convection, if any, which might generate gravity waves.

**Total wave-energy.** To elucidate the effect of thermal tides on gravity waves, we examine wave activity by wave kinetic and potential energy, $E_k$ and $E_p$, averaged over 30 Earth days in a reference frame at fixed local solar times (i.e., composite-mean), which are expressed as follows:

$$\overline{E_k} = \frac{1}{2}\left(\overline{u'^2} + \overline{v'^2}\right) \quad \text{and} \quad \overline{E_p} = \frac{1}{2}\left(\frac{g}{N_a}\right)^2 \frac{\overline{T'^2}}{T_a^2}, \tag{1}$$

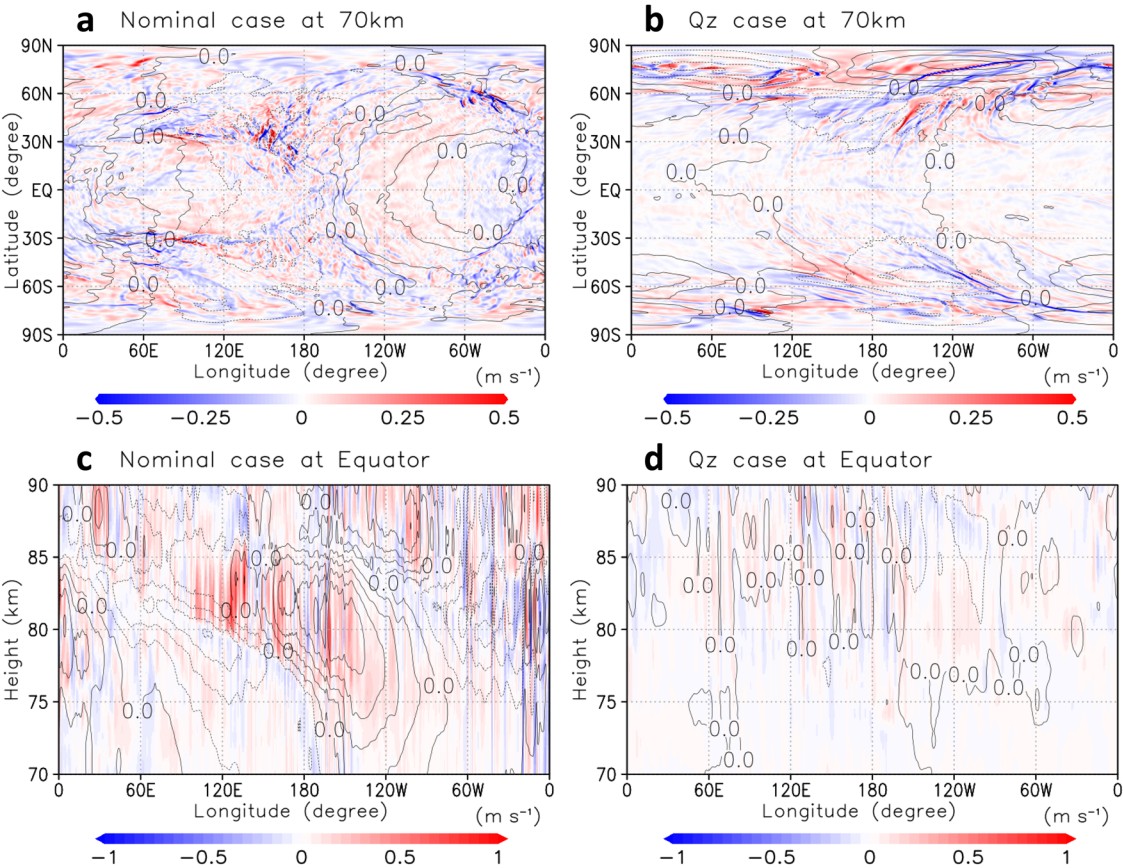

**Fig. 1 Vertical wind velocity (colour, m s$^{-1}$). a**, **b** Longitude–latitude cross-sections at the cloud-top (~70 km) level for nominal and Qz cases, respectively. **c**, **d** Longitude–height cross-sections at the equator for nominal and Qz cases, respectively. Geopotential height disturbances from its zonal average (black contours; intervals are 1000 m$^2$ s$^{-2}$ and dotted contours indicate negative values) are also shown. The subsolar point is located at (176.8°W, 0°N) for **a** and **c**.

where $u'$, $v'$ and $T'$ are horizontal velocities and temperature deviations from their zonal means, respectively; ($\overline{\phantom{-}}$) indicates the time average in a solar-fixed reference frame; $N_a$ and $T_a$ are the zonal and time averages of Brunt–Väisälä frequency and temperature; $g$ is the gravitational acceleration.

Horizontal distributions of the total wave-energy (TWE), $E_k + E_P$, at the cloud-top level for the nominal and Qz cases are shown in Figs. 2a, b, respectively. For the nominal case, the large TWE correlated with thermal tides: wavenumber 1 and 2 components in the mid-latitudes (30°–60°N/S) and the low-latitudes (30°S–30°N), respectively[29]. Zonally uniform distributions of TWE were also significant in the mid-latitudes, corresponding to baroclinic/barotropic waves that actively develop within ~5 days[24]. For the Qz case, the large TWE was zonally uniform and concentrated in the higher latitudes because of the lack of the thermal tides, and the poleward shift of the mid-latitude jets compared with the nominal case, as mentioned above (see Supplementary Fig. 1b).

The vertical structure of TWE obtained for the nominal case (Fig. 2c) also indicated that the small-scale gravity waves shown in Fig. 1c would be associated with thermal tides. Particularly, the TWE increased locally in jet-exit regions (i.e., the local minimum of zonal flow) due to thermal tides. Meanwhile, for the Qz case, the TWE was strongly suppressed near the equator (Fig. 2b, d), showing that there were almost no small-scale gravity waves or their sources near the equator. These results supported that the small-scale gravity waves observed for the nominal case were related to thermal tides, at least in the low-latitudes. See also Supplementary Figs. 2 and 3 for the individual distributions of $E_k$ and $E_P$ for the nominal and Qz cases, respectively.

As for the generation mechanism of gravity waves from the thermal tides, both the velocity-variation and mountain-wave-like mechanisms[10] at jet-exit regions (i.e., JEREmi waves) seemed to work in the present results. The former was expected in regions close to the local minimum of zonal flow (represented by contours in Fig. 2c), namely, in a tilting region from 0°W/180°E at 75 km to 60°W/120°E at 80 km, and the latter was in regions close to the local maximum of geopotential disturbances (represented by contours in Fig. 1c). Because JEREmi waves are frequently observed and simulated in jet/front systems in the terrestrial atmosphere[2], gravity waves generated from the baroclinic/barotropic wave activities in the mid-latitudes and high-latitudes could also be explained by the same mechanisms. We confirmed that the Richardson number was not less than 0.25 in most of the regions, where gravity waves were generated, suggesting that the shear instability was not the primary source of small-scale gravity waves (Supplementary Fig. 4). The amount of energy of these small-scale gravity waves would be sufficient to contribute to the general circulation and material mixing[11,37] as shown below.

**Lagrangian Rossby number.** The Lagrangian Rossby number[38–40], Ro$^{(L)}$, is frequently used as a diagnostic tool to detect the source of spontaneous gravity waves radiation; it is given by

$$\text{Ro}^{(L)} = \left( \left| \frac{\partial \mathbf{v_H}}{\partial t} + \mathbf{v_H} \cdot \nabla \mathbf{v_H} \right| \Big/ f |\mathbf{v_H}| \right) \approx |\mathbf{v_{ag}}|/|\mathbf{v_H}|, \quad (2)$$

where $\mathbf{v_H} = (u, v)$ is the horizontal velocity vector, $\nabla$ is a gradient

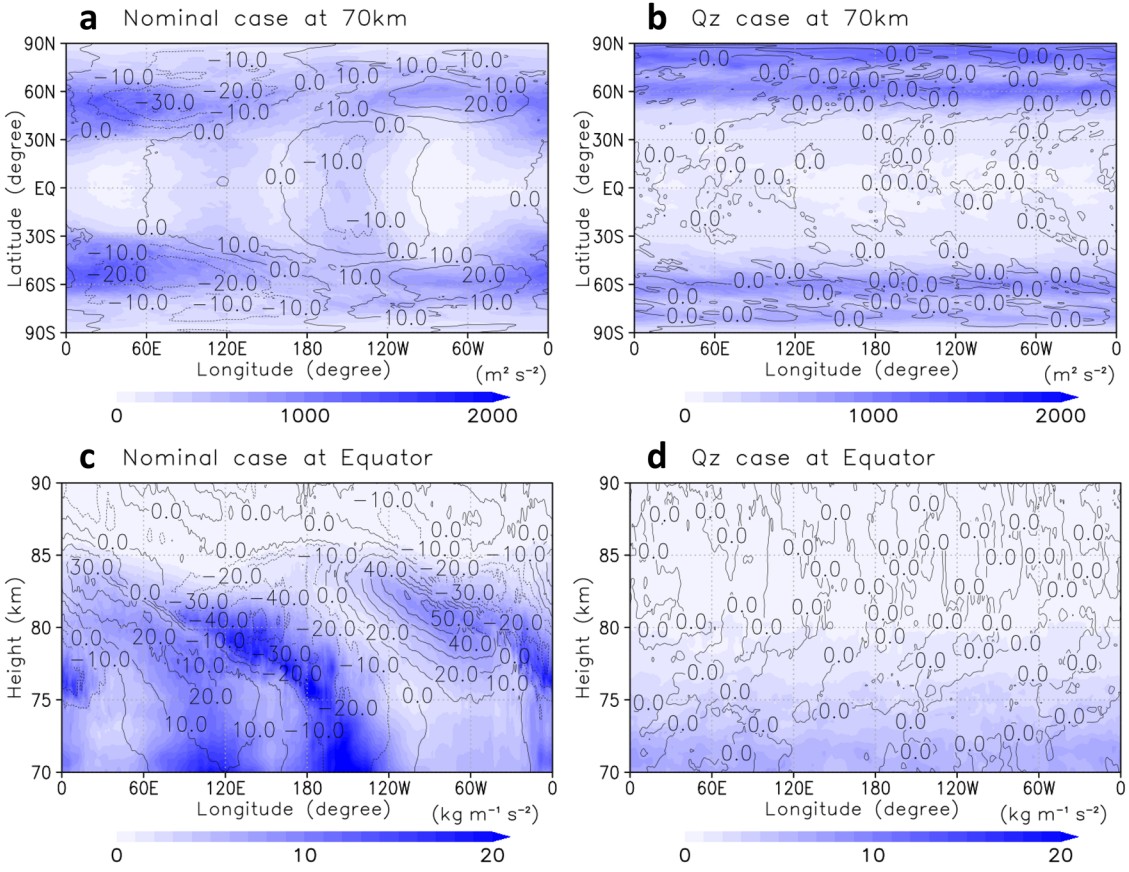

**Fig. 2 Total wave-energy (TWE). a, b** Longitude–latitude cross-sections of TWE (sum of eddy kinetic energy $E_k$ and potential energy $E_p$; colour shade, m$^2$ s$^{-2}$) at the cloud-top (~70 km) level for the nominal and Qz cases, respectively. **c, d** Longitude–height cross-sections of density weighted TWE at the equator (sum of $\rho E_k$ and $\rho E_p$; colour shade, kg m$^{-1}$ s$^{-2}$) for the nominal and Qz cases, respectively. TWE has been averaged over 30 Earth days in a solar-fixed reference frame (i.e., composite-mean). Deviations of zonal flow from its zonal average (black contours; intervals are 10 m s$^{-1}$) are also shown. The subsolar point is located at the centre of panels, (180°E, 0°N), for **a** and **c**.

operator, and $f$ is the Coriolis parameter. Note that Ro$^{(L)}$ is explicitly not related to the horizontal divergence (and vertical velocity) in the flow field but ageostrophic wind, $\mathbf{v}_{ag}$[40]. Although it is difficult to compute Eq. (2) because of the local wind tendency term $\partial \mathbf{v}_H/\partial t$, it has been pointed out that the local wind tendency term is frequently small compared with the advective tendency term[39] $\mathbf{v}_H \cdot \nabla \mathbf{v}_H$. In addition, we confirmed that the local wind tendency term was small and less than approximately 30% of the advective tendency term at most in the present simulation. Note that Ro$^{(L)} \geq 0.5$ was referred to as the high Rossby number regime[40].

To estimate the Coriolis parameter on the slowly rotating Venus planet, we estimate the effective Coriolis parameters, $f_E$, using the zonal mean zonal flow:

$$f_E(\theta, z) = 2 \sin \theta \frac{\overline{u(\theta, z)}}{r \cos \theta}, \qquad (3)$$

where $\theta$ and $z$ are latitude and altitude, respectively; $r = 6052$ km is Venus' radius. In the equatorial region ($\theta \leq 10°$), we fixed the values of $f_E$ at $\theta = 10°$ to extend the meaning of Ro$^{(L)}$ to cover the equatorial latitudes considering equatorial dynamics, where the magnitude of Coriolis term evaluated off the equator represented that of large-scale equatorial disturbances, including thermal tides. Corresponding to this estimate of $f_E$ using the zonal mean zonal flow in Eq. (3), we used $\mathbf{v}_H' = (u', v')$ to estimate Ro$^{(L)}$ in Eq. (2). We removed regions with $|\mathbf{v}_H'| \leq 20$ m/s

so that we could avoid choosing regions with large Ro$^{(L)}$ due to small $|\mathbf{v}_H'|$.

Figures 3a, b show horizontal distributions of Lagrangian Rossby Number, Ro$^{(L)}$, at the cloud-top level calculated for the nominal and Qz cases, respectively. In the low-latitudes, large Ro$^{(L)}$ ($\geq 0.5$) appeared around the jet-exit regions (i.e., the local minimum of zonal flow) and corresponded to the large TWE (Fig. 2a) for the nominal case, whereas there were no regions with large Ro$^{(L)}$ for Qz. The vertical structure of Ro$^{(L)}$ obtained for the nominal case (Fig. 3c) also showed that the regions with large Ro$^{(L)}$ appeared near the local minimum of zonal flow (represented by contours in Fig. 3c), corresponding to the regions with large TWE (Fig. 2c). Moreover, Ro$^{(L)}$ was small everywhere for Qz (Fig. 3d). These results indicated that the gravity waves were spontaneously radiated at the jet-exit regions (i.e., JEREmi waves) produced by the thermal tides in the low-latitudes for the nominal case.

**Vertical momentum flux.** To investigate the roles of small-scale gravity waves on the general circulation, we performed spectral analysis to extract them from thermal tides. Figure 4 shows the vertical momentum flux $\overline{u'w'}$ obtained for different ranges of resolved horizontal scales in longitude–latitude cross-sections at the cloud-top level. Because the thermal tides[29] and baroclinic/barotropic waves[24] consisted of wavenumbers with less than 9 (Fig. 4b), medium-scale and small-scale gravity waves could be extracted by wavenumbers from 10 to 49 (Fig. 4c) and from 50 to

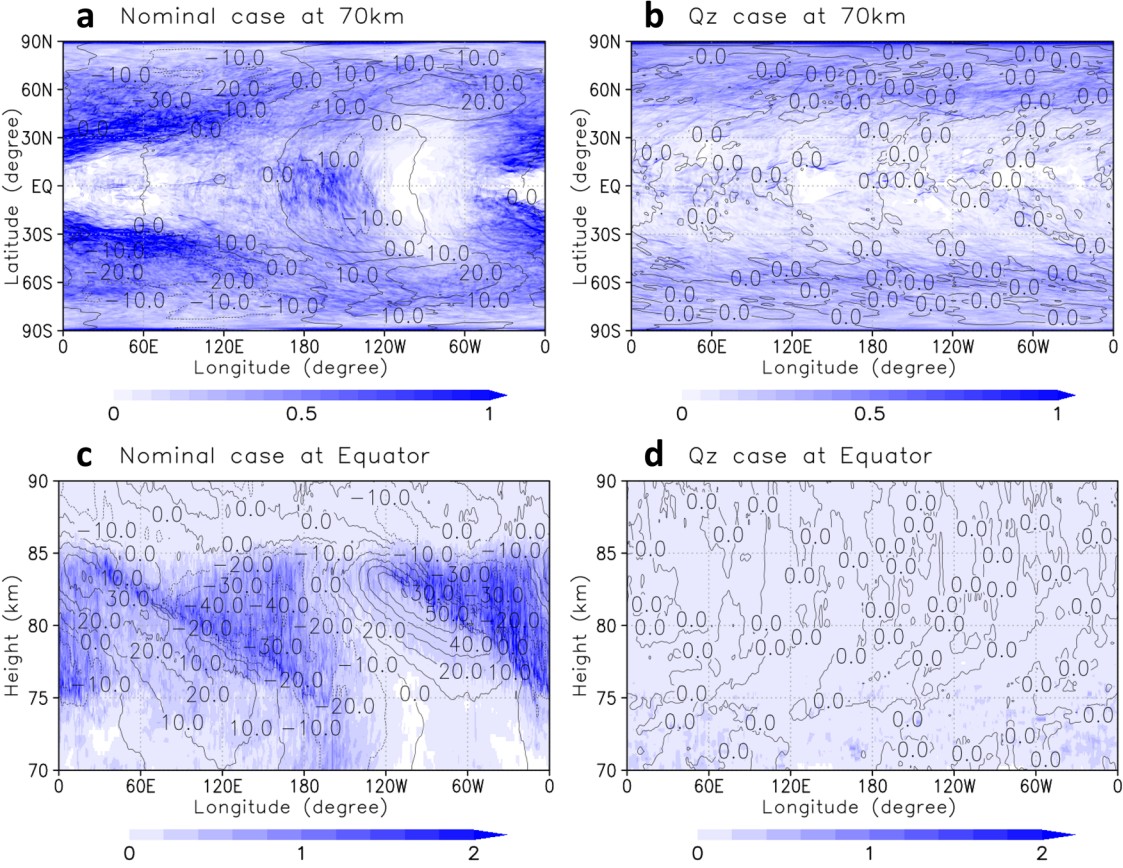

**Fig. 3 Lagrangian Rossby Number (Ro$^{(L)}$; colour shade). a**, **b** Longitude–latitude cross-sections of Ro$^{(L)}$ at the cloud-top (~70 km) level for the nominal and Qz cases, respectively. **c**, **d** Longitude–height cross-sections of Ro$^{(L)}$ at the equator for the nominal and Qz cases, respectively. The values are calculated using the advected tendency term in Eq. (2) with effective Coriolis parameters by Eq. (3) and have been averaged over 30 Earth days in a solar-fixed reference frame (i.e., composite-mean). Atmospheric density normalisation $\rho/\rho_{z=70\,km}$ is multiplied in **c** and **d**. Deviations of zonal flow from its zonal average (black contours; intervals are 10 m s$^{-1}$) are also shown. The subsolar point is located at the centre of panels, (180°E, 0°N), for **a** and **c**.

159 (Fig. 4d), respectively. It was implied from Fig. 4c, d that these gravity waves were spontaneously radiated in the low-latitudes at 120°E–180°E and 60°W–0°W (Fig. 4d) and the mid-latitudes around 60°E/W (Fig. 4c, d). Compared with the Qz case (Supplementary Fig. 5a), wherein the momentum flux was almost zonally uniform, it was suggested that the horizontal structure of vertical momentum flux was strongly affected by the thermal tides (Fig. 4a).

**Acceleration of mean zonal flow.** Figure 5 shows the mean zonal acceleration due to the vertical momentum flux $\overline{u'w'}$ in longitude–height cross-sections at the equator

$$GWD = -\frac{1}{\rho}\frac{\partial\rho\overline{u'w'}}{\partial z} \quad (4)$$

The zonal flow was strongly decelerated along with the jet-exit regions formed by the thermal tides (tilted blue regions in Fig. 5a). As shown in Fig. 5b, these decelerations were mainly due to the large-scale upward propagating thermal tides. The medium-scale gravity waves also contributed to the deceleration in these regions where they were generated spontaneously (Fig. 5c).

The small-scale gravity waves, which were basically generated from jet-exit regions of the thermal tides, appeared above ~76 km and transport momentum flux vertically (Fig. 5d). They decelerated the zonal flow in the region, where they were generated (blue regions) and accelerated the upper zonal flow

locally (e.g., red regions in ~180°E around 80–86 km in Fig. 5d) to compensate for the deceleration due to the thermal tides on average (the same location but in blue in Fig. 5b).

In fact, the acceleration cancelled out about half of the deceleration due to the medium-scale and small-scale gravity waves in 165°E–165°W at ~83 km, which suggested that they would play essential roles in the maintenance of the super-rotation above the cloud layer. Such momentum transport in the low-latitudes was unobserved for the Qz case (Supplementary Fig. 5b).

**Discussion**

We reported that the gravity waves were spontaneously generated from the thermal tides in the Venus atmosphere demonstrated by resolving small-scale gravity waves in a high-resolution general circulation model. At the cloud-top level, gravity waves were generated spontaneously from large-scale atmospheric motions. The main sources are jet-exit regions formed by the thermal tides in the low-latitudes and baroclinically/barotropically unstable regions in the mid-latitudes and high-latitudes. We confirmed that the jet-exit regions had large values of Lagrangian Rossby number, which indicated that both the velocity-variation and mountain-wave-like mechanisms[10] seemed to work for sponta-neous gravity wave generation at jet-exit regions. Momentum flux associated with the gravity waves was significant in the low-lati-tudes, where the thermal tides actively developed in the nominal case. The gravity waves decelerated the zonal flow where they were generated and propagated upward to accelerate the upper

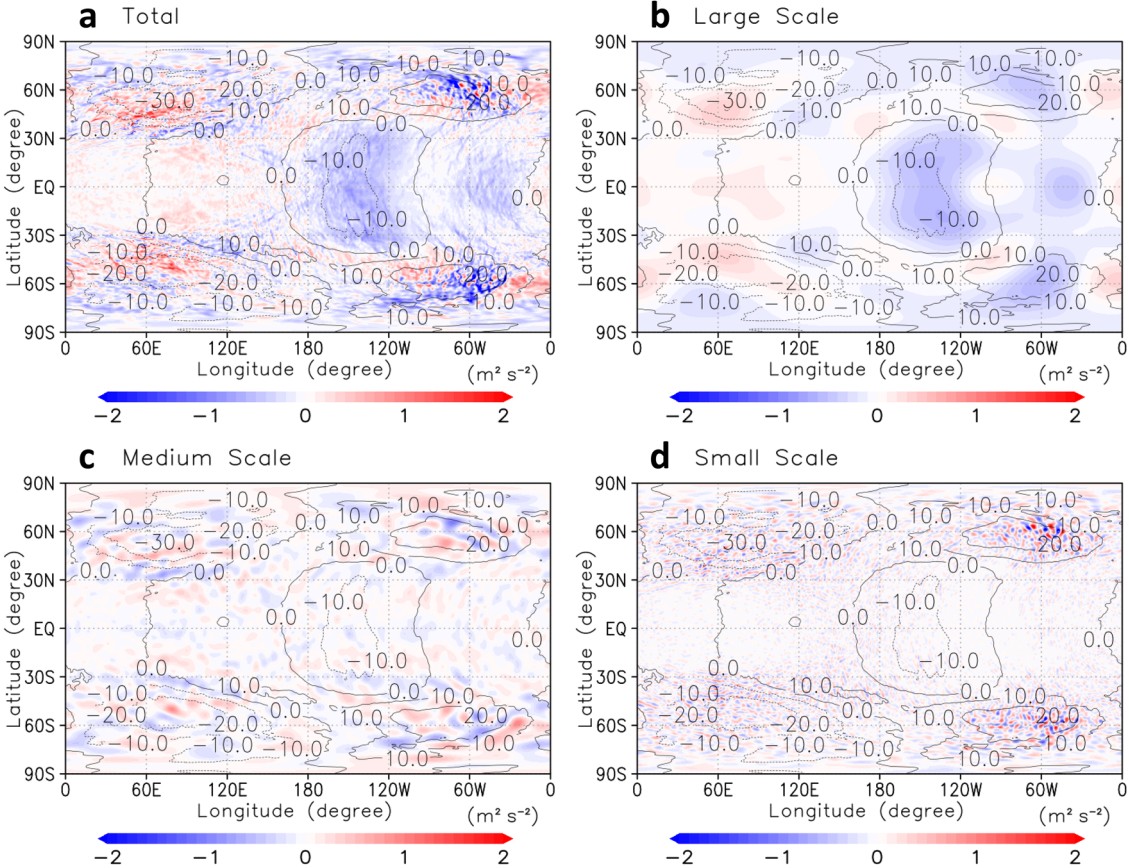

**Fig. 4 Vertical momentum flux. a–d** Longitude–latitude cross-sections of composite-mean vertical momentum flux (m$^2$ s$^{-2}$) at the cloud-top (~70 km) level with different resolved scales, total ($0 \leq k \leq 639$), large-scale ($0 \leq k \leq 9$), medium-scale ($10 \leq k \leq 49$) and small-scale ($50 \leq k \leq 159$) ones, respectively. Here, $k$ denotes a horizontal wavenumber. Zonal flow disturbances from its zonal average (black contours; intervals are 10 m s$^{-1}$) are also shown. The subsolar point is located at the centre of each panel, (180°E, 0°N).

zonal flow locally around 76–86 km. As a result, about half of the deceleration due to the thermal tide could be cancelled out by the gravity waves. The distribution of wave-induced acceleration and deceleration within the thermal tide structure looked, as though the gravity waves were acting to dissipate the semidiurnal tide rather than directly interact with the global super-rotation.

The acceleration/deceleration of the super-rotation induced by waves in the range of 60–70 km altitudes has been estimated in previous studies. Planetary-scale Kelvin waves accelerate at ~0.1–0.3 m/s/day[41–43], whereas planetary-scale Rossby waves decelerate at ~0.15 m/s/day[42,43]. Baroclinic waves accelerate at ~0.05 m/s/day in the mid-latitudes[24]. The thermal tides also accelerate at 1.0 m/s/day or less in the low-latitudes by Akatsuki observations[44] and ~0.2–0.5 m/s/day by another Venus GCM study[45]. Considering that about half of the acceleration due to thermal tides could be cancelled out by gravity waves, the contribution of gravity waves can be comparable to that of planetary-scale waves and more than that of baroclinic waves, although our estimate was rough and not zonally averaged. Therefore, the gravity waves induced by thermal tides play essential roles in the Venus general circulation through the momentum transport, and contribute to material mixing through wave breaking above the cloud layer.

Several observations have suggested that the gravity waves appear in and above the cloud layer[37]. Temperature fluctuations of ~3 K due to gravity waves above the cloud-top (~85 km) observed by radio occultation measurements of the Venus Express[16] and Akatsuki are comparable to those in this study. Although it has been argued that these gravity waves are

generated by cloud layer convection[12–14] and cloud feedback radiative heating[15], our result strongly suggested that spontaneous generation from the thermal tides and baroclinic/barotropic waves is also one of the important sources. It was also suggested by the radio occultation measurements of the Venus Express that the amplitude of unsaturated gravity waves increases with height in the range of 75–90 km compared with 65–80 km[46]. Therefore, small-scale gravity waves in the upper atmosphere (110–140 km) observed by VIRTIS-M[17] are possibly related to spontaneous generation. Since spontaneous generation produces small-scale gravity waves due to wave-capture mechanism[47], especially for large-scale motions varying slowly in time, some small-scale gravity waves at the cloud-top (62–70 km) observed by VMC[18,19] and VIRTIS-M[20] may also be due to the present radiation processes.

Although the Venus rotation is slow (then the Coriolis parameter due to Venus rotation can be ignored), cyclostrophic balance (and thermal wind relation also) has been established in the super-rotation with stable stratification above the cloud layer[48]. Contrary to the terrestrial atmosphere in geostrophic balance (with a small Rossby number), it is expected that the spontaneous gravity wave generation from large-scale motions would be enhanced in the Venus atmosphere as in cases of large Rossby number[7,9]. Actually, the estimate of the Lagrangian Rossby number in the Venus atmosphere tended to be large compared with that in the terrestrial atmosphere. The gravity wave parameterisation in Venus GCMs should be improved in light of the spontaneous generation. Investigating the gravity waves generated by topography and convection, and quantifying the effects of

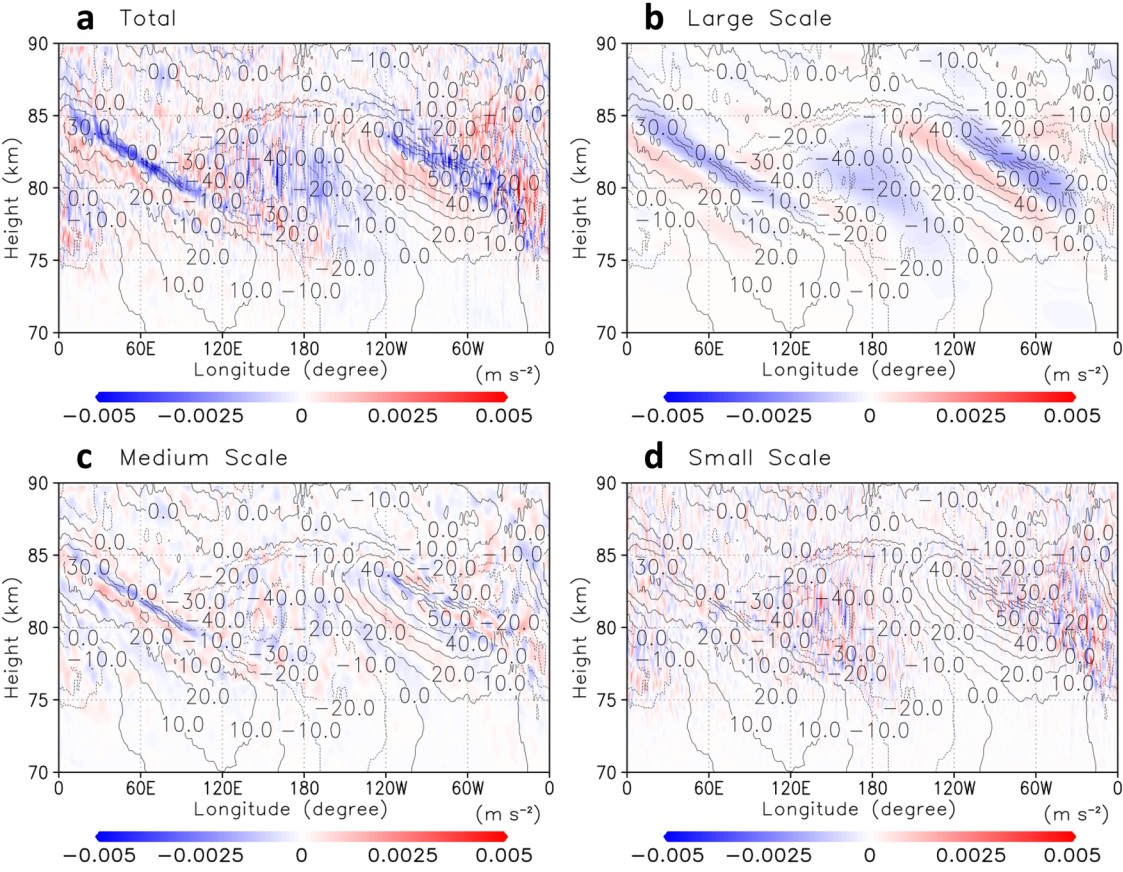

**Fig. 5 Zonal forcing due to vertical momentum flux. a–d** Longitude–height cross-sections of composite-mean zonal forcing (m s$^{-2}$) due to vertical momentum flux expressed by Eq. (4) at the equator with different resolved scales, total ($0 \le k \le 639$), large-scale ($0 \le k \le 9$), medium-scale ($10 \le k \le 49$) and small-scale ($50 \le k \le 159$) ones, respectively. Zonal flow disturbances from its zonal average (black contours; intervals are 10 m s$^{-1}$) are also shown. The subsolar point is located at the centre of each panel, (180°E, 0°N).

gravity waves are crucial future works. More future observations such as radio occultation are also required to determine the source and statistical properties of gravity waves.

## Methods

AFES-Venus is a full nonlinear dynamical GCM in σ coordinate based on the assumption of hydrostatic balance, designed for the Venus atmosphere[24,27]. So far, spontaneous gravity wave generation in the terrestrial atmosphere has mainly been investigated via hydrostatic models[2,9,38,39], because it is unrelated to vertical convection and vertical shear instability. Therefore, we investigated small-scale gravity waves generated at jet-exit regions in the present GCM. The physical processes were simplified for Venus, and there was no topography. The horizontal resolution was T639 (1920 × 960 horizontal grids with ~20 km intervals), which was the highest resolution in GCM simulations of the Venus atmosphere and probably close to the limit of hydrostatic approximation. Note that vertical convection, which, if any, was supposed to be substituted by the dry convective adjustment scheme, possibly began to be explicitly represented in the present model. The atmosphere from the ground to ~120 km was divided into 260 levels in height at non-uniform spacing. The vertical intervals were less than 0.25 km around the cloud layer from 50 to 90 km. The vertical and horizontal eddy diffusion was used in the model. The vertical eddy viscosity coefficient was set to 0.15 m$^2$ s$^{-1}$, and the second-order hyper-viscosity (Laplacian diffusion; $\nabla^4$) was used for the horizontal eddy viscosity whose damping time was ~0.0003 Earth day (~26 s) for the maximum wavenumber component ($k = 639$). This damping time is comparable to the model time step of ~30 s. Rayleigh friction with a relaxation time of 0.5 Earth day was used to represent the surface friction at the lowest level. A sponge layer with coefficients gradually increasing with height was used only for eddy components greater than 80 km. The coefficient of Rayleigh friction $K_R(\sigma)$, are given by the following equation:

$$K_R(\sigma) = \frac{1}{K_R^0}\left[1 + \tanh\left(\frac{z_R \log\sigma - z_R \log\sigma_R}{H_R}\right)\right], \qquad (5)$$

where $K_R^0 = (0.1 \text{ days})^{-1}$, $H_R = 40$ km, $z_R = 80$ km, and $\sigma_R = 5.0 \times 10^{-8}$ are parameters for relaxation time, width, height and sigma level of sponge layer,

respectively. Using these values, the sponge layer acts effectively only greater than ~100 km and at 120 km a relaxation time is 0.05 Earth day (Supplementary Fig. 6).

The atmosphere is driven by the prescribed solar heating based on observations[49], with its maximum ~60 km. Solar heating above 80 km is neglected for numerical stability. The thermal tides of diurnal and semidiurnal (and higher frequency) harmonics, with zonal wave 1 and 2 structures, were excited by the diurnally-varying component of the solar heating (nominal case). In addition, we performed a simulation without thermal tides by excluding the diurnal component of the solar heating (Qz case) to elucidate the effect of thermal tides on small-scale gravity waves. We used Newtonian cooling for the infra-red radiative process; relaxation time was chosen by the work[27,50] and relaxation field is a horizontally uniform temperature prepared by Venus international reference atmosphere (VIRA)[51]. Note that an equator–pole thermal contrast is produced by the solar heating in both the nominal and Qz cases.

An idealised super-rotating flow in solid-body rotation was used for the initial condition. At the equator, zonal velocity increased linearly with height from the ground to 70 km up to 100 m s$^{-1}$ and was constant afterward. The vertical profile of the initial temperature, which included a weakly stratified layer observed in the ranges of 55–60 km[52], was produced by VIRA. The latitudinal distribution of the initial temperature was prepared to be in gradient wind balance with this idealised super-rotating flow. With this initial state, nonlinear numerical simulations were performed for 4 Earth years with T159L260. The simulations were extended for 1 Earth year with T319L260, and 0.5 Earth years with T639L260 to spin up and obtain quasi-equilibrium steady states. We analysed the data for 30 Earth days sampled every Earth day. We set the directions of planetary rotation and basic zonal flow to be eastward (positive).

## Data availability

The datasets generated and/or analysed in the present study are available from the corresponding author upon reasonable request. Source data are provided with this paper.

## Code availability

The model code of the AFES-Venus used in this study is available from the corresponding author upon reasonable request. The GFD-DENNOU Library (http://www.gfd-dennou.org/library/dcl/index.html.en) was used for creating figures.

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

## Acknowledgements

This study was conducted under the Earth Simulator Proposed Research Project with title High Resolution General Circulation Simulation of Venus and Mars Atmosphere using AFES. The work is partly supported by JSPS KAKENHI grants Numbers JP19H01971, JP19H05605, JP20K04062, and JP20K04064. Authors thank to Ms. Hinako Onuma who provided analysed data to produce Supplementay Fig. 4, while the original data was produced in this study.

## Author contributions

N.S. developed the high-resolution version of AFES-Venus with a help of M.T. and performed numerical experiments. N.S. analysed obtained data with a help of Y.F. and H.K. Y.F. created Figs. 1–5 and Supplementary Figs. 1–3 and 5 with a help of N.S. on the platform of GFD-DENNOU Library. N.S. created Supplementary Figs. 4 and 6 on the same platform. All the authors including K.N., T.K., and Y.-Y.H. contributed to the theoretical interpretations.

## Competing interests

The authors declare no competing interests.
