## [Peer Review File · Nature Communications]

Editorial Note:

This peer review file has been amended to remove any third-party material where no permission to publish could be obtained.

Parts of this peer review file have also been redacted as indicated to maintain the confidentiality of unpublished data.

REVIEWER COMMENTS

Reviewer #1 (Remarks to the Author):

This is an important paper in the progress of our understanding of the Venusian atmospheric dynamics. Analyzing high-resolution GCM simulations, this study finds a source of non-orographic gravity waves (GW) in the upper cloud layer. It results in new implications on the momentum budget above the clouds.

While this study is novel, its findings will influence the field, and the manuscript is adequate, I have one major comment that I would like to see addressed before publication. For context, the gravity waves radiated by the topography and the lower cloud convective regions are neglected in these simulations. These sources are generally thought to be the prime origin of GW. GW breaking is parametrized in GCMs simulating the upper atmosphere of Venus and are found to have a sizeable impact on the SS-AS (Subsolar-Antisolar) circulation by breaking above 120 km. One result of this study is the deceleration of the zonal flow below 80 km, and acceleration of the flow above 80 km by GW. However, in the simulations presented here, there is a sponge layer applied on eddy components at altitudes 80 km and above. Hence the question: how does the sponge layer impact these results? More details are needed on the coefficients of the sponge layer, and how they vary with altitude. Sugimoto 2014 does not provide more details on the sponge layer. This comment boils down to one remark that would greatly benefit the article if it were addressed: For what physical reason do GW emitted by the lower cloud convective region reach 120 km in other studies, while GW emitted in the upper cloud deck do not reach higher than 86 km in this study? If the sponge layer were modified, would the conclusions of this study be different? Additionally, is there a SS-AS circulation in the simulations? If not, is it canceled out by the sponge layer?

Other comments:

There is an abuse of quotes in the abstract.

Abstract: thorough → through

Also in the Venus atmosphere, [...] → [...] are also expected to play important roles in the Venusian atmosphere.

What does Qz mean exactly? Q is for insolation, what is z for?

Figure 4 caption: typo in “small-scale(5”

Lower cloud layer (60 km) → 50 km

Reviewer #2 (Remarks to the Author):

This paper describes the first results from a set of very high resolution numerical simulations of a Venus-like atmosphere (the highest published to date), focusing on the generation of small-scale gravity wave features at levels around and above those of the visible clouds in the Venus atmosphere. The simulations make a number of simplifying assumptions and approximations in their boundary conditions and representation of physical processes, but employ a fully non-linear solver for the resolved dynamics at extremely high spatial resolution (20 km in the horizontal) which is unprecedented in the context of Venus. The results confirm that relatively small-scale (200-2000 km wavelength?) gravity waves are generated spontaneously in various places within the circulation, with some similarities to those observed in the Venus atmosphere, though the authors make no detailed attempt to compare the simulated waves directly with observations. How do the wavelengths of GWs generated in simulations and their preferred locations, for example, compare with GWs observed on Venus? The simulations further demonstrate (convincingly, though perhaps unsurprisingly) that a strong semi-diurnal thermal tide modifies the distribution of gravity wave activity, especially at low latitudes.

As a first demonstration of the generation of relatively small-scale gravity wave activity within a fully 3D time dependent simulation of a Venus-like atmosphere, this is an interesting and worthwhile study, though it does leave a number of significant questions unaddressed. The model simulations themselves are reasonably well documented in the paper, with some good illustrations and movies of the waves in the supplementary material. The discussion of the mechanisms generating the small-scale waves within the simulations is, however, rather too simplistic and not very convincing or insightful. The formulation of the model (as a hydrostatic code with convective adjustment and no surface topography) specifically excludes certain mechanisms previously suggested as possible sources of gravity waves (e.g. see Lefevre et al. JGR, 122 (2016) 134-149), as also acknowledged by the authors themselves. So in the absence of a more careful comparison with observations, the reader is left wondering whether the gravity waves produced in this model occur for the same physical reasons as on Venus itself. The authors compute the magnitude of typical accelerations of the flow by the generation and dissipation of the small-scale gravity waves, though it is not entirely clear from the discussion in the paper in its present form just how significant these waves are e.g. for the large-scale global super-rotation of Venus's atmosphere.

Nevertheless, the simulations are impressive in scale and detail and represent a first significant step towards modelling the global circulation of the Venus atmosphere at mesoscale resolution without relying on ad hoc GW parameterizations. But the authors should provide some more convincing

discussion of the sources of the waves in their simulation, and ideally also make some more careful comparison with observations of Venus, before the paper is accepted. Some more detailed comments follow.

Lines 21-22 and elsewhere: The authors refer to “spontaneous gravity wave radiation” from nearly “balanced flows” here and in several other places. But for a slowly rotating planet it is not entirely clear what this means. Is the Lighthill-Ford spontaneous adjustment process implied here? If so, this really needs more justification, especially in the context of a non-geostrophic large-scale flow. If this is not what is intended, it should be clarified.

Line 62: Please define what is meant by QZ where this first appears (otherwise it seems a bit cryptic and mysterious).

Line 75: Here it is claimed that the gravity waves in the model are resolved, but without stating explicitly either what the model resolution is or what are the wavelengths of the majority of the gravity waves. This would be the place to state this to justify this statement.

Line 94: Does “vertical flow” here mean vertical velocity?

Line 129: Presumably “Total wave-energy” is shown by the color shading in Fig. 2? Please clarify. Also note typo “Erath” on line 133.

Lines 138-151: The discussion here of source mechanisms for GWs is too simplistic and superficial. It is not enough simply to have either local divergence or uplift over isentropic surfaces to generate gravity waves (as opposed to any other kind of wave) - especially on scales much smaller than that of the large-scale flow. The local Rossby number or timescale of forcing, for example, are also relevant to determine whether gravity waves would result - has this been investigated? Ref. 2 is cited and discusses mechanisms such as spontaneous adjustment or “unbalanced (geostrophic) instabilities”. Have the authors considered the possibility of local instabilities generating the GWs in their simulations? The regions of origin in Fig. 4, for example, look like regions of strong local vertical shear - which might suggest the possibility of Kelvin-Helmholtz shear instability if the Richardson number was small enough. Why are short wavelength GWs generated rather than waves comparable in size to the tides or BWs? Are they related to spontaneous adjustment or local instabilities? What are the quantitative criteria for either mechanism being invoked? And where is the evidence that these criteria are satisfied?

Line 144: Does the flow have to be “zonal”? I don’t think so....

Line 147: “sown” [sic]

Lines 175-192 and Fig. 4: The distribution of wave-induced acceleration and deceleration within the thermal tide structure looks as though the GWs are acting to dissipate the semi-diurnal tide, rather than interact directly with the global super-rotation - discuss?

Lines 234-5: Is a hydrostatic model well suited to the task of simulating interactions with gravity waves of such small scale? This should at least be justified.

Line 242: Does “second order hyper-viscosity” actually mean a classic Laplacian diffusion?

Line 243: 0.0003 Earth days is around 25s - is this the model timestep?

Lines 252-3: If the relaxation field is horizontally uniform, how can it drive the large-scale circulation? There must surely be an equator-pole thermal contrast.....please explain.

Reviewer #3 (Remarks to the Author):

Gravity wave radiation from thermal tides in the Venus atmosphere simulated by an ultra-high-resolution GCM by Norihiko Sugimoto et al.

This manuscript describes results from a very high resolution Venus general climate model simulations that is able to explicitly resolve the small scale gravity waves that have been previously proposed to possibly play a significant role in the momentum balance that maintains the observed super-rotation of the Venus atmosphere. While the impact of parameterized waves has been

investigated in atmospheric models, the realistic generation of such waves has been limited. One source of gravity waves is from convective motions within the cloud deck of Venus. Another potentially significant source for gravity wave is the adjustment process associated with an almost-balanced vortex structure, which would require a sufficiently high resolution global scale model for investigation. This mechanism is the focus of this paper. The main result is that gravity waves generated within the jet-exit regions formed by the thermal tides in the tropics and by barotropically/baroclinically unstable regions at mid-to-high latitudes are important sources of gravity waves, and should be considered in further developing parameterizations for gravity wave activity in the Venusian atmosphere. I find that the conclusion is well supported by the modeling and would expect that the result would be of significant interest to modelers of the Venusian atmosphere. I imagine that the paper could be suitable for publication in Nature Communications, though I am inclined to think that it is perhaps too specialized and limited in scope. While the thermal tide is clearly responsible for generating gravity wave activity, I don't have any clear idea of how this process actually works. I would prefer to see a deeper analysis of this mechanism, which would likely necessitate publication elsewhere.

In any event, I offer the following comments that might help to improve the manuscript. A general comment is that some of the material discussed in the paragraph starting at line 138 would be more effectively stated in the opening paragraphs of the manuscript to better motivate the current study.

Comments.

Line 25: "...gravity waves propagate <vertically> and

Line 27: "It is strongly suggested...." I assume it is shown in the manuscript that the 3-D structure is affected.

Line 31: In the terrestrial atmosphere, orographically forced waves and non-orographic waves associated with convection (particularly in the tropics) account for much of the gravity wave spectrum.

Line 59: This is an overly long sentence.

Line 89: Remove "to" in "...are shifted to poleward..." and add "are" to "...and stronger..."

Line 138: The first half of this paragraph ought to appear in the introductory material as part of the motivation for the study.

Line 147: “ ... <shown> by contours....”

Line 150: It is stated that the amount of energy in these small-scale gravity waves is enough to contribute to the general circulation and material mixing. How is this demonstrated?

The authors use the conditional construction “would be” very frequently. I think it would be better to use “is” or “are” instead.

Figure 1 caption. Better to state that “vertical velocity” is shown, rather than “vertical flows”.

Figures 2c and 2d evidently show TWE multiplied by density. This should be stated in the caption, as it is for figures S6 and S7. I now see that this scaling has been applied, though it has been awkwardly stated.

The caption for Figure 8 should state that the two panels are for the no-tide simulation. I think it is worth restating that Figure S8a is the counterpart to Figure 3a and Figure S8b is the counterpart to Figure 4a in the main text.

Line 247: I assume that the tides are excited by the diurnally-varying component of solar heating, allowing excitation of diurnal and semidiurnal (and higher frequency) harmonics, with zonal wave 1 and 2 structure, as is apparent in Figure 4 and the supplementary animations.

Supplementary Material:

Line 72: “...large E_k and E_p ...” Do you mean large scale?

Line 74: What does it mean that “ E_k and E_p are kept quite low in low latitudes” Presumably this is due to the absence of a source, which would be the tide.

Line 75: “It is strongly suggested that the would be generated by the thermal tides” These results strongly suggest that ..ARE generated by the thermal tides

Reviewer #1 (Remarks to the Author):

This is an important paper in the progress of our understanding of the Venusian atmospheric dynamics. Analyzing high-resolution GCM simulations, this study finds a source a non-orographic gravity waves (GW) in the upper cloud layer. It results on new implications on the momentum budget above the clouds.

While this study is novel, its findings will influence the field, and the manuscript is adequate, I have one major comment that I would like to see addressed before publication. For context, the gravity waves radiated by the topography and the lower cloud convective regions are neglected in these simulations. These sources are generally thought to be the prime origin of GW. GW breaking is parametrized in GCMs simulating the upper atmosphere of Venus and are found to have a sizeable impact on the SS-AS (Subsolar-Antisolar) circulation by breaking above 120 km. One result of this study is the deceleration of the zonal flow below 80km, and acceleration of the flow above 80km by GW. However, in the simulations presented here, there is a sponge layer applied on eddy components at altitudes 80km and above. Hence the question: how does the sponge layer impact these results? More details are needed on the coefficients of the sponge layer, and how they vary with altitude. Sugimoto 2014 does not provide more details on the sponge layer. This comment boils down to one remark that would greatly benefit the article if it were addressed: For what physical reason do GW emitted by the lower cloud convective region reach 120km in other studies, while GW emitted in the upper cloud deck do not reach higher than 86km in this study? If the sponge layer were modified, would the conclusions of this study be different? Additionally, is there a SS-AS circulation in the simulations? If not, is it canceled out by the sponge layer?

Answer to reviewer #1:

Thank you very much for carefully reading our manuscript and the valuable suggestions. We have revised the manuscript following Reviewers' comments. Especially, we are grateful for your comment regarding with the sponge layer. We think that model considerations would be more complete in the revised manuscript.

We have added the equation (Eq.(5) in p.26) and figure of the sponge layer (Fig.S9 in supplementary information) in the revised manuscript. It varies with height and acts effectively above ~100 km (see also below figure). We have checked dependency on the coefficient of the sponge layer in the previous studies (Sugimoto et al. 2014JGR) and confirmed that there is no significant change in general circulation with different coefficient. Therefore, the conclusions of this study would be robust as long as focusing on the phenomena below ~100 km.

As indicated by the recent results of Akatsuki radio occultation measurement, gravity waves radiated from the lower cloud convective region cannot propagate to above the cloud top level of ~70 km (see also answer for the reviewer #2). We have shown that some of the gravity waves dissipate around 86 km due to the breaking, but other and secondary gravity waves will propagate toward 120 km and above. Though this is beyond the scope of the present study, gravity waves generated from thermal tides may contribute to the general circulation including SS-AS circulation in the upper atmosphere.

Regarding with SS-AS circulation, we have found some part of such circulation around 80 km in the previous study (Takagi et al., 2018; see also figure 5 of <https://doi.org/10.1002/2017JE005449> & figure 6 of <https://doi.org/10.1002/2017JE005449>). However, SS-AS circulation above 120 km cannot be focused on in the present model (AFES-Venus) because of the upper limit of the model (120 km) with the sponge layer acting effectively above ~100 km. We have added some discussions in the revised manuscript (1.310-324).

[Redacted]

[redacted]

Other comments:

There is an abuse of quotes in the abstract.

We have corrected them in the revised manuscript.

Abstract: thorough → through

We have corrected it.

Also in the Venus atmosphere, [...] → [...] are also expected to play important roles in the Venusian atmosphere.

We have corrected it following a suggestion.

What does Qz mean exactly? Q is for insolation, what is z for?

Sorry for not explaining the meaning. We have added explanation for Qz in the revised manuscript (1.83-84).

Figure 4 caption: typo in “small-scale(5”

We have corrected it.

Lower cloud layer (60 km) → 50 km

We have corrected it.

Reviewer #2 (Remarks to the Author):

This paper describes the first results from a set of very high resolution numerical simulations of a Venus-like atmosphere (the highest published to date), focusing on the generation of small-scale gravity wave features at levels around and above those of the visible clouds in the Venus atmosphere. The simulations make a number of simplifying assumptions and approximations in their boundary conditions and representation of physical processes, but employ a fully non-linear solver for the resolved dynamics at extremely high spatial resolution (20 km in the horizontal) which is unprecedented in the context of Venus. The results confirm that relatively small-scale (200-2000 km wavelength?) gravity waves are generated spontaneously in various places within the circulation, with some similarities to those observed in the Venus atmosphere, though the authors make no detailed attempt to compare the simulated waves directly with observations. How do the wavelengths of GWs generated in simulations and their preferred locations, for example, compare with GWs observed on Venus? The simulations further demonstrate (convincingly, though perhaps unsurprisingly) that a strong semi-diurnal thermal tide modifies the distribution of gravity wave activity, especially at low latitudes.

As a first demonstration of the generation of relatively small-scale gravity wave activity within a fully 3D time dependent simulation of a Venus-like atmosphere, this is an interesting and worthwhile study, though it does leave a number of significant questions unaddressed. The model simulations themselves are reasonably well documented in the paper, with some good illustrations and movies of the waves in the supplementary material. The discussion of the mechanisms generating the small-scale waves within the simulations is, however, rather too simplistic and not very convincing or insightful. The formulation of the model (as a hydrostatic code with convective adjustment and no surface topography) specifically excludes certain mechanisms previously suggested as possible sources of gravity waves (e.g. see Lefevre et al. JGR, 122 (2016) 134-149), as also acknowledged by the authors themselves. So in the absence of a more careful comparison with observations, the reader is left wondering whether the gravity waves produced in this model occur for the same physical reasons as on Venus itself. The authors compute the magnitude of typical accelerations of the flow by the generation and dissipation of the small-scale gravity waves, though it is not entirely clear from the discussion in the paper in its present form just how significant these waves are e.g. for the large-scale global super-rotation of Venus's atmosphere.

Nevertheless, the simulations are impressive in scale and detail and represent a first significant step towards modelling the global circulation of the Venus atmosphere at mesoscale resolution without relying on ad hoc GW parameterizations. But the authors should provide some more convincing discussion of the sources of the waves in their simulation, and ideally also make some more careful comparison with observations of Venus, before the paper is accepted. Some more detailed comments

follow.

Answer to reviewer #2:

Thank you very much for carefully reading our manuscript and the valuable suggestions. We have revised the manuscript following reviewers' comments. Especially, we are grateful for your comments regarding with discussion of the mechanisms and comparison with observations. Adding these elements, we think that the revised manuscript would be convincing and of more significance.

We agree that the mechanism of gravity wave generation is very important topic. However, since the mechanism of spontaneous gravity wave radiation in the terrestrial planet was well explained by Yasuda et al. (2015), we would not focus on the detail mechanism in the present study. It is very difficult to specify the source of gravity waves even for the terrestrial planet. Alternatively, we have added new section of Lagrangian Rossby number to diagnose spontaneous gravity wave radiation at jet-exit regions (1.184-228 & Fig.3). We also mentioned value of Richardson number so that the shear instability is not the source of small-scale gravity waves at least in the present results (1.178-181). Dedicated studies will be needed in comparison with other gravity wave sources, such as topography, vertical convection and KH instability.

In addition, we have added several typical estimates for acceleration/deceleration of the super-rotation (in 60-70 km altitudes) caused by waves, thermal tides $\sim 0.2-1.0$ m/s/day (Horinouchi et al., 2020; Yamamoto et al., 2021), planetary-scale Kelvin waves $\sim 0.1-0.3$ m/s/day (Del Genio & Rossow, 1990; Imamura, 2006; Kouyama et al., 2015) and Rossby waves ~ 0.15 m/s/day (Imamura, 2006; Kouyama et al., 2015), and baroclinic waves ~ 0.05 m/s/day (Sugimoto et al., 2014GRL), in order to compare them with those by gravity waves in the present study (1.297-309). Although present estimate is rough and not zonally averaged, considering that about a half of the deceleration due to the thermal tide could be cancelled out by the gravity waves, contribution by gravity waves can be comparable to that by planetary-scale waves and more than that by baroclinic waves.

Regarding with observations, first it is suggested by the Venus Express radio occultation measurement that gravity waves are radiated above the cloud top (Tellmann et al., 2012; see figure 5 of <https://doi.org/10.1016/j.icarus.2012.08.023>). It is also suggested that amplitude of unsaturated gravity waves grows with height in 75-90 km compared with 65-80 km (Ando et al., 2015).

[redacted]

Here, temperature fluctuation due to gravity waves exists in the cloud layer (60 to 72 km) in latitudes from 45N(S) to 75N(S), which will be caused by the convection in the cloud layer. However, these gravity waves do not propagate to the upper atmosphere because there is no temperature fluctuation from 72 to 80 km. Nevertheless, it is clearly seen that temperature fluctuation exists again above the cloud top (80 to 85 km) in the mid-latitudes. These results strongly suggested that there is another source of gravity waves above the cloud top different from cloud level convection. Note that since the Venus Express was the polar orbit, there are few observations in the low-latitudes.

[redacted]

Second, above figure shows 27 temperature disturbances with high pass filtered (for vertical wavelength less than 4 km) in the low-latitudes from 30S to 30N observed by radio occultation measurements of the Akatsuki Venus Climate Orbiter. Again it is suggested that gravity waves

would be generated from convections in the cloud layer (50 to 60 km) and saturate around the cloud top level (~70 km). In addition to this, there are gravity waves above the cloud top (higher than 75 km), and those amplitudes increase with altitude, again suggesting another source of gravity waves.

[redacted]

It is also suggested that gravity waves with vertical wave lengths less than 4 km are extracted in the low-latitudes and high altitude. See figures 4.2 and 4.8 of Ryota Mori, (2020), "Properties of gravity wave packets detected in radio occultation temperature profiles of the Venus atmosphere", pp.49. Department of Complexity Science and Engineering, Tokyo University. The figure shows wavelet analysis of temperature fluctuations observed by the radio occultation measurements of both Venus Express and Akatsuki orbiter.

Because these results have not been published yet, we cannot add these figures and/or references. Alternatively, we have explained small-scale gravity waves observed above the cloud top by the radio occultation as much as possible in the revised manuscript. Although future work is necessary, temperature fluctuations of ~3 K caused by gravity waves above the cloud top (~85 km) observed by radio occultation measurements of the Venus Express and Akatsuki are comparable to those in the present studies (l.311-313).

[redacted]

We have also added several references of small-scale gravity waves observed by the Venus Express and discussed them as much as possible.

Figure 3 of <https://doi.org/10.1029/2008JE003073> provides the results of Visible and Infrared Thermal Imaging Spectrometer-Mapper (VIRTIS-M) onboard the Venus Express. Small-scale gravity waves whose horizontal wave lengths are from 90 to 400 km are observed in the upper atmosphere from 110 to 140 km altitude range (Garcia et al., 2009). Some of these small-scale gravity waves might be radiated from the thermal tides.

At the cloud tops (62 to 70 km altitude), many types of small-scale gravity waves whose horizontal wave lengths are tens of km are also obtained with the Venus Monitoring Camera (VMC) (Markiewicz et al., 2007; Piccialli et al., 2014) and VIRTIS-M (Peralta et al., 2008) onboard the Venus. As shown from figure 6 from <https://doi.org/10.1016/j.icarus.2013.09.012>

These small-scale gravity waves may be produced by wave capture mechanism (Bühler and McIntyre, 2005) at jet-exit regions, especially for large-scale motions varying slowly in time. We have added these comparison with observations in the revised manuscript (1.48-59 & 1.310-324).

Lines 21-22 and elsewhere: The authors refer to “spontaneous gravity wave radiation” from nearly “balanced flows” here and in several other places. But for a slowly rotating planet it is not entirely clear what this means. Is the Lighthill-Ford spontaneous adjustment process implied here? If so, this really needs more justification, especially in the context of a non-geostrophic large-scale flow. If this is not what is intended, it should be clarified.

Yes, in the previous version, we have used the term “spontaneous gravity wave radiation” as the same meaning of “Lighthill-Ford spontaneous adjustment process”. However, Lighthill-Ford spontaneous adjustment process is mainly used in the context of shallow water system. Therefore, in the present study, we think that it is better to use JEREmi (Jet Exit Region Emitted) waves as defined by Plougonven and Zhang (2014). We have explained these terminology more detail in the revised manuscript (1.30-44).

Though the slowly rotating planet of Venus, cyclostrophic balance (and thermal wind relation) has been established in the super rotation with stable stratification around the cloud layer (Sanchez-Laverge et al., 2017). Therefore, large-scale motions are mostly balanced and spontaneous gravity waves radiation from nearly balanced similar to the terrestrial planet would be occur, while it could be enhanced due to large Rossby number. We have added these explanations in the revised manuscript (1.325-332).

Line 62: Please define what is meant by QZ where this first appears (otherwise it seems a bit cryptic and mysterious).

Sorry for not explaining the meaning. We have added explanation for Q_z in the revised manuscript (1.83-84).

Line 75: Here it is claimed that the gravity waves in the model are resolved, but without stating explicitly either what the model resolution is or what are the wavelengths of the majority of the gravity waves. This would be the place to state this to justify this statement.

Thank you for an important suggestion. We have explained them in the revised manuscript (1.99-101).

Line 94: Does “vertical flow” here mean vertical velocity?

We have corrected it.

Line 129: Presumably “Total wave-energy” is shown by the color shading in Fig. 2? Please clarify. Also note typo “Erath” on line 133.

We have explained colour shade and corrected typo.

Lines 138-151: The discussion here of source mechanisms for GWs is too simplistic and superficial. It is not enough simply to have either local divergence or uplift over isentropic surfaces to generate gravity waves (as opposed to any other kind of wave) - especially on scales much smaller than that of the large-scale flow. The local Rossby number or timescale of forcing, for example, are also relevant to determine whether gravity waves would result - has this been investigated? Ref. 2 is cited and discusses mechanisms such as spontaneous adjustment or “unbalanced (geostrophic) instabilities”. Have the authors considered the possibility of local instabilities generating the GWs in their simulations? The regions of origin in Fig. 4, for example, look like regions of strong local vertical shear - which might suggest the possibility of Kelvin-Helmholtz shear instability if the Richardson number was small enough. Why are short wavelength GWs generated rather than waves comparable in size to the tides or BWs? Are they related to spontaneous adjustment or local instabilities? What are the quantitative criteria for either mechanism being invoked? And where is the evidence that these criteria are satisfied?

Thank you for very important comments. We have investigated local (Lagrangian) Rossby number and Richardson number. It is confirmed that regions with large local (Lagrangian) Rossby number located at jet-exit regions (see Fig.3), which is the quantitative criteria for spontaneous gravity wave radiation (JEREmi waves). It is also confirmed that unbalanced instabilities due to vertical shear cannot happens (see below figures also). Small-scale gravity waves are signature of wave capture mechanism (Bühler & McIntyre, 2005) caused by spontaneous gravity wave radiation at jet-exit regions (JEREmi waves), especially for large-scale motions varying slowly in time.

Above figures are static stability (left) and Richardson number (right) in longitude-height

cross-sections at the equator. Regions with small static stability (~80km, 120-180E) correspond to those with small Richardson number. However, Regions with large temperature fluctuations (contour) caused by the small-scale gravity waves do not correspond to those with small static stability and small Richardson number, suggesting that small scale gravity waves are not generated from unbalanced instability (Kelvin-Helmholtz shear instability) at least in the present results (l.178-181).

Line 144: Does the flow have to be “zonal”? I don’t think so....

We have corrected it.

Line 147: “sown”[sic]

We have corrected it.

Lines 175-192 and Fig. 4: The distribution of wave-induced acceleration and deceleration within the thermal tide structure looks as though the GWs are acting to dissipate the semi-diurnal tide, rather than interact directly with the global super-rotation -discuss?

This is what we want to mention. We have added some discussions in the revised manuscript (l.294-296).

Lines 234-5: Is a hydrostatic model well suited to the task of simulating interactions with gravity waves of such small scale? This should at least be justified.

So far, spontaneous gravity wave radiation in the terrestrial atmosphere has been mainly investigated by a hydrostatic model because it is not related to vertical convection and vertical shear instability. Therefore, we can investigate small-scale gravity waves generated at jet-exit regions in the present GCM. We have added these explanations in the revised manuscript (l.341-345).

Line 242: Does “second order hyper-viscosity” actually mean a classic Laplacian diffusion?

Yes, we have explained it explicitly, “second-order hyper-viscosity (Laplacian diffusion; ∇^4)” (l.354-357).

Line 243: 0.0003 Earth days is around 25s - is this the model timestep?

We have explained model time step of ~30s in the revised manuscript. Therefore, the maximum

wavenumber component is damped at each time step (l.354-358).

Lines 252-3: If the relaxation field is horizontally uniform, how can it drive the large-scale circulation? There must surely be an equator-pole thermal contrast.....please explain.

Since the solar heating is not horizontally uniform, it produces an equator-pole thermal contrast. We have added this explanation (l.375-377).

Reviewer #3 (Remarks to the Author):

Gravity wave radiation from thermal tides in the Venus atmosphere simulated by an ultra-high-resolution GCM by Norihiko Sugimoto et al.

This manuscript describes results from a very high resolution Venus general climate model simulations that is able to explicitly resolve the small scale gravity waves that have been previously proposed to possibly play a significant role in the momentum balance that maintains the observed super-rotation of the Venus atmosphere. While the impact of parameterized waves has been investigated in atmospheric models, the realistic generation of such waves has been limited. One source of gravity waves is from convective motions within the cloud deck of Venus. Another potentially significant source for gravity wave is the adjustment process associated with an almost-balanced vortex structure, which would require a sufficiently high resolution global scale model for investigation. This mechanism is the focus of this paper. The main result is that gravity waves generated within the jet-exit regions formed by the thermal tides in the tropics and by barotropically/baroclinically unstable regions at mid-to-high latitudes are important sources of gravity waves, and should be considered in further developing parameterizations for gravity wave activity in the Venusian atmosphere. I find that the conclusion is well supported by the modeling and would expect that the result would be of significant interest to modelers of the Venusian atmosphere. I imagine that the paper could be suitable for publication in Nature Communications, though I am inclined to think that it is perhaps too specialized and limited in scope. While the thermal tide is clearly responsible for generating gravity wave activity, I don't have any clear idea of how this process actually works. I would prefer to see a deeper analysis of this mechanism, which would likely necessitate publication elsewhere.

In any event, I offer the following comments that might help to improve the manuscript. A general comment is that some of the material discussed in the paragraph starting at line 138 would be more effectively stated in the opening paragraphs of the manuscript to better motivate the current study.

Answer to reviewer #3:

Thank you very much for carefully reading our manuscript and the valuable suggestions. We have revised the manuscript following Reviewers' comments. Especially, we are grateful for your comment regarding with discussion of the mechanisms. We have explained and discussed the mechanism in more detail, adding new section of Lagrangian Rossby number in order to diagnose spontaneous gravity wave radiation (l.184-228 & Fig.3). We think that the revised manuscript would be convincing and of more significance.

Comments.

Line 25: "...gravity waves propagate <vertically> and

We have corrected it.

Line 27: "It is strongly suggested...." I assume it is shown in the manuscript that the 3-D structure is affected.

We have modified this sentence.

Line 31: In the terrestrial atmosphere, orographically forced waves and non-orographic waves associated with convection (particularly in the tropics) account for much of the gravity wave spectrum.

Thank you for an important comment. We have modified the sentence.

Line 59: This is an overly long sentence.

We have separated it into two sentences.

Line 89: Remove "to" in "...are shifted to poleward..." and add "are" to "...and stronger..."

We have corrected them.

Line 138: The first half of this paragraph ought to appear in the introductory material as part of the motivation for the study.

Thank you for an important suggestion. We have moved several sentences to the introduction.

Line 147: " ... <shown> by contours...."

We have corrected it.

Line 150: It is stated that the amount of energy in these small-scale gravity waves is enough to contribute to the general circulation and material mixing. How is this demonstrated?

The authors use the conditional construction "would be" very frequently. I think it would be better to use "is" or "are" instead.

We have added several typical estimates for acceleration/deceleration of the super-rotation (in 60-70 km altitudes) caused by waves in order to compare them with those by gravity waves in the present study (1.297-309). Although present estimate is rough and not zonally averaged, considering that about a half of the deceleration due to the thermal tide could be cancelled out by the gravity waves, contribution by gravity waves can be comparable to that by planetary-scale waves and more than that by baroclinic waves. In addition, we use “is” or “are” instead of the conditional construction “would be” as much as possible.

Figure 1 caption. Better to state that “vertical velocity” is shown, rather than “vertical flows”.

We have corrected it.

Figures 2c and 2d evidently show TWE multiplied by density. This should be stated in the caption, as it is for figures S6 and S7. I now see that this scaling has been applied, though it has been awkwardly stated.

We have corrected it.

The caption for Figure 8 should state that the two panels are for the no-tide simulation. I think it is worth restating that Figure S8a is the counterpart to Figure 3a and Figure S8b is the counterpart to Figure 4a in the main text.

Thank you for good suggestions. We have added these explanations.

Line 247: I assume that the tides are excited by the diurnally-varying component of solar heating, allowing excitation of diurnal and semidiurnal (and higher frequency) harmonics, with zonal wave 1 and 2 structure, as is apparent in Figure 4 and the supplementary animations.

You are right. We have added these explanations.

Supplementary Material:

Line 72: “...large Ek and Ep...” Do you mean large scale?

We have corrected it, “large amount of Ek and Ep”.

Line 74: What does it mean that “Ek and Ep are kept quite low in low latitudes” Presumably this is

due to the absence of a source, which would be the tide.

We have corrected this sentence.

Line 75: "It is strong suggested that the would be generated by the thermal tides" These results strongly suggest that ..ARE generated by the thermal tides

We have corrected this sentence.

REVIEWERS' COMMENTS

Reviewer #1 (Remarks to the Author):

Thank you for your answers, the revised version provides more details and analysis, improving the manuscript and supporting the conclusions of this work. I find it suitable for publication.

Reviewer #2 (Remarks to the Author):

In producing their revised manuscript the authors have now largely addressed the issues I raised. The resulting manuscript is now a novel and interesting paper on a topical issue that demonstrates some stimulating analogies between Venus's atmosphere and the Earth's. The English will need some attention at the copy editing stage, but I am now content to recommend acceptance of the paper in more or less its present form.

Reviewer #3 (Remarks to the Author):

I am satisfied with the authors' response to my review review in particular, and to comments by the two other reviewers. I would note that the grammar could be improved, but it is not an impediment for readability.

R. John Wilson

Reviewer #1 (Remarks to the Author):

Thank you for your answers, the revised version provides more details and analysis, improving the manuscript and supporting the conclusions of this work. I find it suitable for publication.

Answer to reviewer #1:

Thank you very much for carefully reading our manuscript again. Following other Reviewers' comments, we have revised the manuscript mainly in English, asking English language editing service.

Reviewer #2 (Remarks to the Author):

In producing their revised manuscript the authors have now largely addressed the issues I raised. The resulting manuscript is now a novel and interesting paper on a topical issue that demonstrates some stimulating analogies between Venus's atmosphere and the Earth's. The English will need some attention at the copy editing stage, but I am now content to recommend acceptance of the paper in more or less its present form.

Answer to reviewer #2:

Thank you very much for carefully reading our manuscript again. Following your comment, we have revised the manuscript mainly in English, asking English language editing service.

Reviewer #3 (Remarks to the Author):

I am satisfied with the authors' response to my review review in particular, and to comments by the two other reviewers. I would note that the grammar could be improved, but it is not an impediment for readability.

R. John Wilson

Thank you very much for carefully reading our manuscript again. Following your comment, we have revised the manuscript mainly in English, asking English language editing service.